# Defining a positive work environment for hospital healthcare professionals: A Delphi study

Susanne M. Maassen[1]*, Catharina van Oostveen[2,3], Hester Vermeulen[4,5], Anne Marie Weggelaar[3]

1 Department of Quality and Patient Care, Erasmus MC University Medical Center, Rotterdam, The Netherlands, 2 Spaarne Gasthuis Hospital, Haarlem, The Netherlands, 3 Erasmus School of Health, Policy and Management, Erasmus University, Rotterdam, The Netherlands, 4 Department IQ Healthcare, Radboudumc University Medical Center, Nijmegen, The Netherlands, 5 Faculty of Health and Social Studies, HAN University of Applied Sciences, Nijmegen, The Netherlands

* s.maassen@erasmusmc.nl

## Abstract

### Introduction

The work environment of healthcare professionals is important for good patient care and is receiving increasing attention in scientific research. A clear and unambiguous understanding of a positive work environment, as perceived by healthcare professionals, is crucial for gaining systematic objective insights into the work environment. The aim of this study was to gain consensus on the concept of a positive work environment in the hospital.

### Methods

This was a three-round Delphi study to establish consensus on what defines a positive work environment. A literature review and 17 semi-structured interviews with experts (transcribed and analyzed by open and thematic coding) were used to generate items for the Delphi study.

### Results

The literature review revealed 228 aspects that were clustered into 48 work environment elements, 38 of which were mentioned in the interviews also. After three Delphi rounds, 36 elements were regarded as belonging to a positive work environment in the hospital.

### Discussion

The work environment is a broad concept with several perspectives. Although all 36 elements are considered important for a positive work environment, they have different perspectives. Mapping the included elements revealed that no one work environment measurement tool includes all the elements.

**Data Availability Statement:** All relevant data are within the manuscript and its Supporting Information files.

**Funding:** The research was funded by the Citrien Fonds of ZonMW (grant 8392010042) and conducted on behalf of the Dutch Federation of University Medical Centers' Quality Steering program: https://www.sturenopkwaliteit.nl The funders had no role in study design, data collection and analysis, decision to publish, or preparation of the manuscript.

**Competing interests:** The authors have declared that no competing interests exist.

## Conclusion

We identified 36 elements that are important for a positive work environment. This knowledge can be used to select the right measurement tool or to develop interventions for improving the work environment. However, the different perspectives of the work environment should be considered.

## Introduction

A positive work environment (WE) for healthcare professionals is important for good patient care [1]; it reduces hospital-acquired infection rates [2–4], hospital mortality [5], re-admissions [6], and adverse events [2, 3]. Furthermore, a positive WE is strongly associated with attracting and retaining healthcare professionals [7, 8], which is crucial in times of healthcare staff shortages, especially with regard to the COVID-19 pandemic. Therefore, the WE is receiving increasing attention in scientific research, with over 1.1 million hits on Google Scholar and almost 500,000 publications in the last five years.

The WE is a complex concept with several perspectives. Damschroder, Aron [9] define the WE as the inner setting of the organization where staff interact with the organization within which they work [9]. Others have added four WE contexts to this definition [4, 9]. The first is the task context, which includes the work that needs to be performed, clarity of the role, and the workload. For the nurses' WE, this has been defined as [10] *'the organizational characteristics of a work setting that enable or constrain professional nursing practice'* and includes nursing foundations in quality of care, nurses' participation in hospital policy, staffing and resources adequacy, and collegial nurse–physician relationships. The second WE context is the social context, which includes relations, interaction between employees, and teamwork [4, 9]. A concept used to reflect the social context is e.g. the civility climate, described as '*shared perception of the extent to which an organization rewards, supports, and expects a) respect and acceptance, b) cooperation, c) supportive relationships between coworkers, and d) fair conflict resolution*' [11, 12]. The third context is the physical context, which involves work safety, working conditions, labor environments, housing, and the physical and mental health of employees. Research has highlighted the impact this has on the constraints and complaints of employees, such as burnout [7] and the need for more sick leave [13]. The fourth context is organizational culture, which involves the values, norms, and culture [4, 11] and has been defined as 'the way we do things around here' [14]. Research on organizational culture usually focuses on specific aspects, not the whole concept. For instance, safety culture is studied by researching aspects connected to the clinical setting, such as patient safety and learning from adverse events [11, 14], whereas organizational culture considers the whole organization, including administration, technicians, and logistics, which have not been well studied so far [15].

Research has shown that achieving a positive WE is challenging for professionals working together in interprofessional teams, departments, organizations, and organization networks [16]. Because a positive WE is important for patients and employees, many organizations have embarked on efforts to measure their WE. However, there are many instruments for measuring the WE and these might lack consensus on which elements are important for a positive WE [17, 18]. A clear and unambiguous understanding of the most important aspects for a positive WE, as perceived by healthcare professionals, is crucial if hospitals want to gain systematic objective insights into the WE. The purpose of this study was to gain consensus on the concept of a positive WE and to determine which elements define a positive WE in hospitals.

## Methods

### Study design

A three-round Delphi study was conducted to identify elements of a positive WE. The Delphi study is a group facilitation technique with an iterative multi-stage process designed to transform individual opinions into group consensus [19, 20]. The Delphi technique provides the opportunity to involve individuals with diverse expertise and from several locations and backgrounds through a digital survey [20]. Because the WE has gained worldwide attention, we were able to involve international experts. We started by generating items followed by three Delphi study rounds (Fig 1). CREDES (recommendations for Conducting and REporting of DElphi Studies) [21] was used to design and report the study.

### Generation of items

Items pertaining to healthcare professionals' WE were generated from a literature review [18] and semi-structured expert interviews.

We searched Embase, Medline Ovid, Web of Science, Cochrane CENTRAL, CINAHL EBSCOhost, and Google Scholar for literature on instruments used to measure perceptions of healthcare professionals about their hospital WE. This search identified 6397 papers. The four criteria for inclusion were: 1) written in English, 2) reporting an original WE measurement instrument for healthcare professionals in hospitals; 3) not a translation or adaptation of another instrument; and 4) description of psychometric properties and distributed items into factors for construct validity. Based on these criteria, 37 papers were eligible for inclusion (see Maassen, Weggelaar-Jansen [18] for more information). Next, we extracted the items (subscales) of the instruments. After extraction, three researchers sorted the items, discussed the clustering into 48 (potential) elements of WE, and agreed on a description of each element (S1 Data).

To ensure current opinions of WE were measured by the WE instruments, we conducted semi-structured expert interviews. The aim was to validate the elements found in the literature search. Participants responsible for WE in their organization were recruited from various backgrounds by convenience sampling (Table 1). One researcher (SM) interviewed participants either in person (n = 15) or by video call (n = 2). The interviews began with an open

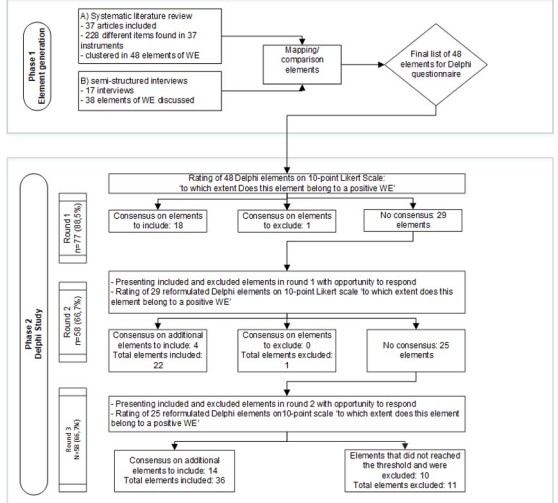

**Fig 1. Delphi flowchart.**

Table 1. Demographics of interview participants.

|  | N |
|---|---|
| Gender |  |
| Female | 9 |
| Male | 8 |
| Organization |  |
| University hospital | 9 |
| Teaching hospital | 5 |
| Research institute | 3 |
| Occupation |  |
| Board member | 5 |
| Head of a medical department | 3 |
| Professor with expertise in work environment | 5 |
| Nurse manager | 1 |
| Director of quality | 1 |
| Patient safety officer | 1 |
| Nurse liaison officer | 1 |

defining question: *which topics do you believe have a positive or negative influence on healthcare professionals' WE*? Next, respondents were asked to illustrate their view with examples of a WE. Interviews were conducted until data saturation was reached. The interviews were transcribed verbatim and analyzed using open and axial coding by one researcher (SM) to identify WE items (S2 Data). Two researchers (CO, AW) checked half of the analysis each.

Then, three researchers (AW, SM, CO) compared and discussed the WE elements derived from the interviews and literature search until consensus was achieved on the final list of elements and descriptions for the Delphi survey.

## Delphi study

**Selection of participants and ethics.** To obtain a solid understanding of a positive healthcare professionals' WE in hospitals, we consulted an international, interdisciplinary Delphi panel with expertise in the WE or with practical experience in steering WE in hospitals. The following participants were selected:

1. corresponding authors of papers found in the literature search; 9/36 invited authors (25%) agreed to participate;

2. experts participating in the interviews; 9/17 respondents (52%) agreed to participate;

3. hospital board members, medical board members, nursing board members, human resource managers, quality officers, or head nurses of hospitals working in Dutch research collaborations on quality and safety; 70/105 (67%) agreed to participate (Table 2).

Ethics approval was not necessary under Dutch law as no patient data were collected. Following the European Union General Data Protection Regulation, all potential participants first received an email inviting them to participate. Only those who gave informed consent to participate received the anonymous online Delphi survey. All collected data were anonymized and thus confidential.

**Data collection.** A Delphi study should have at least two rounds so participants can give feedback and revise previous responses [22]. This study used three rounds so participants could provide feedback and reconsider responses, thereby preventing respondent fatigue [22].

**Table 2. Characteristics of Delphi participants.**

| | Round 1 | | Round 2 | | Round 3 | |
|---|---|---|---|---|---|---|
| **Age** | **Mean** | **SD** | **Mean** | **SD** | **Mean** | **SD** |
| | 47.6 | 10.9 | 47.9 | 11.8 | 48.7 | 10.5 |
| **Organization** | **n** | **%** | **n** | **%** | **N** | **%** |
| University hospital | 51 | 66.2 | 41 | 70.7 | 41 | 70.7 |
| Teaching hospital | 18 | 23.4 | 12 | 20.7 | 11 | 19.0 |
| Research institute | 7 | 9.1 | 4 | 6.9 | 5 | 8.6 |
| Private company | 1 | 1.3 | 1 | 1.7 | 1 | 1.7 |
| **Occupation** | **n** | **%** | **n** | **%** | **N** | **%** |
| Nurse | 17 | 22.1 | 14 | 24.1 | 11 | 19.0 |
| Medical specialist / department head | 6 | 7.8 | 3 | 5.2 | 5 | 8.6 |
| Board member | 9 | 11.7 | 6 | 10.3 | 8 | 13.8 |
| Manager | 18 | 23.4 | 11 | 19.0 | 10 | 17.2 |
| Human resources manager | 1 | 1.3 | 1 | 1.7 | 1 | 1.7 |
| Researcher | 10 | 13.0 | 10 | 17.2 | 9 | 15.5 |
| Director / executive | 11 | 14.3 | 9 | 15.5 | 9 | 15.5 |
| Quality advisors | 5 | 6.5 | 4 | 6.9 | 5 | 8.6 |

Based on the literature review and interviews, the research team agreed an English description for each of the 48 elements to be used in the survey. The first draft of the questionnaire was pilot tested on content, flow, and clarity by two independent hospital policy advisors and the research team. The final first-round Delphi survey was administered digitally via LimeSurvey, an online survey web app (https://www.limesurvey.com). Two reminder emails were sent with an interval of seven days to non-responding participants. Participants were asked to rate to which extent an element belonged to the concept of positive WE using a 10-point scale ranging from one (not at all) to ten (totally). The 10-point rating scale is a commonly known rating scale for Delphi participants and widely used in Delphi studies [20, 23]. A score of 8–10 was considered an agreement.

In the second round, all elements with consensus following the forward set threshold in the first round (see section 'Data analysis and consensus' for consensus method and thresholds) were presented to the participants to give them the opportunity to provide feedback. The remaining elements and their reformulated descriptions were resubmitted to the participants. Again, they were asked to assess the extent to which each item belongs to the concept of a positive WE on the same scale (from 'not at all' to 'totally'). Participants were invited to provide feedback on all the elements to help the researchers reformulate the elements for round three. The same procedure, question, and rating scale were also used for the third and final Delphi round.

**Data analysis and consensus.** For this study, we defined consensus as a percentage of agreement on 'element belongs to a positive WE' [20]. Two thresholds for consensus were applied. The threshold for inclusion in the first and second rounds was set at 80%, indicating that >80% of the participants rated the element eight or higher. This threshold is slightly higher than the median Delphi threshold recommended by Diamond, Grant [20]. All elements scoring exactly 80% were presented again in the next round. An exclusion threshold was agreed if >50% of the participants rated an element seven or less. The research team reformulated all the remaining elements based on the respondents' feedback. Elements scoring just below the exclusion threshold were evaluated by the research team and if there were reasons to believe that misjudgment was likely, the team discussed whether the element should be

included in the next round [20]. For the last round, the threshold was set at 70%, according to Diamond's recommendation, indicating that >70% of the participants rated the element eight or higher [20]. All elements that did not reach this threshold were excluded. The included elements were compared with the list of elements that were initially extracted by WE measurement tools in the literature review [18].

## Results

### Generation of items

The literature review revealed 228 aspects that were clustered into 48 WE elements (S1 Data). Thirty-eight of these elements were further discussed by the 17 interview participants (Table 3 and S2 Data). The research team gave each element a description based on the literature review and interviews.

The WE elements mentioned most frequently in the interviews were presence of a supportive manager (n = 10), leadership (n = 9), autonomy (n = 7), supportive coworkers (n = 7), teamwork (n = 7), and structural and electronical resources available (n = 7). These elements were also frequently used in the WE measurement instruments studied in the literature. One exception was 'job satisfaction'; this was often found in literature, but the respondents did not mention it.

### Delphi study

In total, 88 participants received the Delphi questionnaires. The response rates in rounds 1, 2, and 3 were 88.9%, 66.7%, and 66.7%, respectively. Most respondents were employed in a university hospital during all three rounds and the mean age was 48 years (Table 2).

After the first round, consensus was reached for 18 elements with percentages ranging from 82% to 95% (Table 3). Three elements ('professionalism and competency', 'conflict management', and 'employees as valuable partners') exactly reached the 80% threshold for consensus and were presented again in round two. The element 'participation in policy making', which reached 50% consensus on exclusion, was reformulated. 'Rewards' and 'performance measurement' both scored below the 50% threshold. As a result, 'Rewards' was excluded but 'performance measurement' was reformulated and presented again in round 2 because the research team doubted the accuracy of the element description.

In the second round, 25/58 participants (43%) commented on the list of elements that reached consensus in round 1. All respondents recognized and acknowledged the list, although some questions arose about the unique identity of some elements. One respondent noted:

*'This is a good list. I recommend considering how some of these are connected. Consider if they are truly unique constructs'* (participant A).

After round two, four elements reached consensus: respect (82%), supportive coworkers (82%), supportive manager (82%), and supportive organizational atmosphere (81%). No element reached the exclusion threshold. Hence, the research team reformulated 25 elements based on the respondents' comments. This resulted in almost identical descriptions of the elements 'level of stress' and 'workload', so we decided to merge both elements into 'workload' (Table 3).

In the third round, 17/58 respondents (29%) commented on the list of elements that reached the threshold for consensus in round two. Although respondents recognized the elements on the list, concerns were raised about overlapping elements.

**Table 3. Overview of results from the Delphi rounds.**

| Elements | Number of times described in 37 articles | Number of times mentioned in 17 interviews | Consensus reached in round | % score $\geq$ 8 | Description with consensus |
|---|---|---|---|---|---|
| Autonomy | 10 | 7 | 1 | 95% | Autonomy in accomplishing tasks and making decisions (within own work area) |
| Career advancement | 2 | 1 | 1 | 92% | Promotion and career development are possible in the organization |
| Challenging and fun work | 2 | 3 | 1 | 92% | Having fun and being challenged at work |
| Control over practice setting | 7 | 3 | 1 | 91% | Each profession has sufficient professional status in the organization to influence others and to deploy resources when required |
| Employees as valuable partners | 2 | 0 | 1 | 90% | Employees are viewed as valuable partners and therefore represent value for the organization |
| Feeling valued | 4 | 4 | 1 | 90% | Others give you the feeling that your efforts and contributions are valuable |
| Internal work motivation | 2 | 2 | 1 | 87% | Intrinsic work motivation of employees |
| Job satisfaction | 9 | 0 | 1 | 87% | Satisfaction with the content of the work, salary, and secondary employment conditions |
| Leadership | 6 | 9 | 1 | 87% | The actions of formal and informal leaders in an organization (or unit) to influence change and facilitate excellence in practice |
| Multidisciplinary collaboration | 13 | 6 | 1 | 87% | Good multidisciplinary relations and collaboration based on mutual respect and trust |
| Open communication | 9 | 6 | 1 | 87% | Equal and open (blame-free) communication and feedback between professionals and between different organizational levels |
| Patient-centered culture | 3 | 2 | 1 | 86% | The care is tailored to the patient and their loved ones' values, preferences, and needs |
| Personal development | 2 | 2 | 1 | 85% | Opportunities for personal growth |
| Physical comfort | 2 | 3 | 1 | 85% | Healthy physical work conditions |
| Professionalism and competency* | 6 | 4 | 1 | 84% | Collaboration with colleagues who act professionally and competently |
| Professional development | 9 | 6 | 1 | 83% | Opportunities for developing competencies, expertise, and skills needed to improve performance in a profession |
| Relational atmosphere | 8 | 5 | 1 | 83% | An atmosphere generated by certain behaviors and interpersonal relationships supporting the team spirit. An encouraging, welcoming environment based on mutual respect and trust |
| Safety climate | 2 | 6 | 1 | 82% | The perception of a safety culture |
| Respect | 3 | 1 | 2 | 82% | Colleagues appreciate and accept each other as they are |
| Supportive coworkers | 2 | 7 | 2 | 82% | Positive helpful behavior between colleagues |
| Supportive manager | 15 | 10 | 2 | 82% | Managers who support and coach professionals |
| Supportive organizational atmosphere | 2 | 3 | 2 | 81% | The organization is aware of the unique interests of various professionals and acts accordingly |
| Teamwork | 12 | 7 | 3 | 81% | Respectful, effective, and efficient cooperation on a common purpose |
| Trust | 3 | 3 | 3 | 80% | The extent to which employees trust the organization, their supervisors, and their coworkers |
| Working conditions | 2 | 0 | 3 | 80% | Employees can provide quality with the available time and resources |
| Celebrating achievements | 1 | 0 | 3 | 78% | Milestones are celebrated |
| Cultural values | 4 | 4 | 3 | 78% | Attitude and behavior: the way that 'we do things' in the organization and work units |
| Information distribution | 6 | 0 | 3 | 77% | Distribution of information to employees |

(*Continued*)

**Table 3.** (Continued)

| Elements | Number of times described in 37 articles | Number of times mentioned in 17 interviews | Consensus reached in round | % score ≥ 8 | Description with consensus |
|---|---|---|---|---|---|
| Innovation and change readiness | 6 | 4 | 3 | 77% | Willingness to change, innovate, and improve (the care process) |
| Organizational learning | 2 | 4 | 3 | 77% | Focus on organizational learning and quality improvement |
| Role clarity | 3 | 2 | 3 | 75% | Clear description of responsibilities and competences needed for the job |
| Scheduling | 4 | 2 | 3 | 75% | Work schedules match the work-life balance |
| Self-care | 1 | 0 | 3 | 71% | Focus on health and wellbeing of employees |
| Shared mission and vision | 5 | 4 | 3 | 71% | Clear mission and vision with tasks aligned accordingly |
| Staffing adequacy | 7 | 2 | 3 | 70% | The number of personnel is aligned to the work that needs to be done |
| Workload | 10 | 3 | 3 | 70% | The balance between the workload experienced by the employee and the workload imposed by the company |
| Conflict management | 6 | 0 | Exclusion | 69% | Conflicts within the organization are resolved |
| Adequate authorization and clear chain of command | 4 | 0 | Exclusion | 68% | A clear chain of command and decision-making procedure |
| Structural and electronical resources | 6 | 7 | Exclusion | 64% | Availability of structural material and electronical resources for the job |
| Job retainment | 3 | 1 | Exclusion | 63% | Attracting and retaining employees |
| Justice | 1 | 0 | Exclusion | 60% | Righteousness, equitableness, and moral decisions |
| Task orientation | 4 | 1 | Exclusion | 60% | Employees know which tasks are expected from them |
| Incident reporting and handling of errors | 2 | 1 | Exclusion | 59% | A system for reporting and analyzing incidents is available |
| Performance measurement | 4 | 1 | Exclusion | 52% | Performance quality and patient outcomes are measured |
| Rewards | 3 | 1 | R1 | 49% | The balance between the amount of work that needs to be done and the mental and physical burden it creates |
| Participation in policy making | | 1 | R3 | 49% | Employees can participate in policy development and decision-making at the organizational level |
| Working according to guidelines | 1 | 1 | R3 | 34% | Use of professional standards and guidelines |
| Level of stress = merged with workload in R3 | 1 | 0 | R2 | | A good balance between the workload experienced by an employee and the workload imposed by the company |

´This is a good list. Some items are at least very interconnected or perhaps similar. For example, aren't autonomy and control over practice setting similar?' (participant B).

The final Delphi round led to consensus on inclusion for 14 additional elements, ranging from 70% to 81% consensus > 8 (Table 3). The ten remaining elements did not achieve the threshold of 70% for consensus and were excluded.

Mapping the 36 elements that reached consensus and the elements extracted from the measurement tools derived from the literature review [18] showed that between 2 and 14 elements are included in the WE instruments (Table 4) and nine measurement tools included 11 or more elements [15, 24–31]. The ten elements that did not reach consensus were frequently used in WE measurement tools, especially 'conflict management' and 'participation and policy making'.

## Discussion

The purpose of this study was to gain consensus on the concept of a positive WE by describing which elements comprise a positive WE in hospitals. We identified 48 elements based on a

**Table 4. Cross table of elements X measurement tool.**

The table lists authors (rows) against work context elements (columns), marked with X where present.

Column headers (work context elements):
Relational atmosphere; Multidisciplinary collaboration; Supportive coworkers; Teamwork; Respect; Open communication; Trust; Feeling valued; Autonomy; Control over practice setting; Leadership; Supportive manager; Supportive organizational atmosphere; Celebrating achievements; Conflict management; Adequate authorization and clear chain of command; Justice; Task orientation; Structural and Electronical Resources available; Working conditions; Staffing adequacy; Workload; Scheduling; Professional development; Career development; Professionalism & competency; Personal development; Rewards; Job satisfaction; Level of stress; Self-care; Physical comfort; Role clarity; Internal work motivation; Challenging & fun work; Participation in policy making at organizational level; Shared mission, vision; Performance measurement; Job retainment; Employees as valuable partners; Cultural values; Patient-centered culture; Organizational learning; Innovation and change readiness; Information distribution; Working according to guidelines; Incident reporting & handling of errors; Safety climate; Total nr of elements in instrument; Total of elements with consensus present.

Authors (rows):
Abraham and Foley*; Adams, Bond*; Aiken and Patrician*; Appel, Schuler*; Berndt, Parsons*; Bonneterre, Ehlinger*; Clark, Sattler*; Duddle and Boughton*; Erickson, Duffy*; Erickson, Duffy*; Estabrooks, Squires*; Flint, Farrugia*; Friedberg, Rodriguez*; Gagnon, Paquet*; Ives-Erickson, Duffy*; Ives-Erickson, Duffy*; Jansson von Vultée*; Kalisch, Lee*; Kennerly, Yap*; Klingle, Burgoon*; Kobuse, Morishima*; Kramer and Schmalenberg*; Lake*; Li, Lake*; Mays, Hrabe*; McCusker, Dendukuri*; McSherry and Pearce*; Pena-Suarez, Muniz*; Rafferty, Philippou*; Reid, Courtney*; Saillour-Glenisson, Domecq*; Schroder, Medves*; Siedlecki and Hixson*; Stahl, Schirmer*; Upenieks, Lee*; Whitley and Putzier*; Wienand, Cinotti*.

Total nr of elements in instrument (per author): 13, 10, 7, 13, 11, 7, 18, 8, 9, 9, 11, 9, 9, 13, 9, 12, 3, 11, 6, 7, 10, 12, 11, 7, 6, 11, 7, 8, 13, 8, 16, 11, 4, 10, 7, 9, 10.

Total of elements with consensus present (per author): 11, 8, 7, 11, 9, 7, 14, 7, 8, 8, 9, 9, 6, 12, 8, 11, 2, 11, 5, 5, 8, 11, 9, 7, 5, 10, 6, 6, 11, 8, 11, 10, 4, 9, 6, 8, 8.

* Full reference details are available in Maassen, Weggelaar-Jansen [18] and in S1 Data

literature review and 17 interviews. Of the 48 elements, 36 were confirmed by experts in the Delphi study. One element ('rewards') was excluded by experts in the first round, but was included in three measurement tools in the literature [28, 32, 33]. Ten elements were dropped because they did not reach the threshold in the final round. Our mapping showed that no WE measurement tool included every element. This may be due to the length of the tool. Some tools are rather short and include fewer elements, e.g., those developed by Siedlecki and Hixson [34] (13 items addressing four elements), Kennerly, Yap [35] (22 items addressing six elements), and Mays, Hrabe [36] (12 items addressing six elements).

As the Delphi participants indicated, some elements are similar, e.g., 'scheduling' and 'staffing adequacy' or 'feeling valued' and 'employee as valuable partners'. The differences between them are based on perspective; the first is the employees' perspective and the second is a more organizational perspective. Rugulies [8] distinguished the different perspectives according to the macro, meso, and micro level of WE. The macro level includes economic, social, and political structures; the meso level involves workplace structures and psychosocial working conditions; and the micro level includes individual experiences and cognitive and emotional processes [8]. We included meso- and micro-level elements in our WE elements because both levels are paramount in improving the WE for healthcare professionals. According to Rugulies [8], interventions to influence the WE are best done at the meso level where the employer can have an influence. However, input for these interventions comes from the micro level [37]. Which interventions are effective for which elements and how they work still remains unclear [16, 38].

Our descriptions distinguish between concrete elements and broad abstract elements of the WE. For instance, 'control over practice setting' versus 'autonomy'. Damschroder, Aron [9] used a broad definition for the WE that included all the elements found in our own Delphi study. Lee, Stone [4] further described four WE contexts: 1) task, 2) social, 3) physical, and 4) cultural. However, some of our elements belonged to multiple contexts, deriving from different research domains and perspectives. One example is 'working conditions'–this element concerns how fit the environment is for the employees' physical health and development, i.e., the physical context. On the other hand, this element also concerns how much time and how many resources employees can provide for their task, i.e., the task context.

Harrison, Henriksen [39] described the healthcare WE based on a sociotechnical system approach. The healthcare WE has three components: organization, personnel, and outcomes. Interactions between the organization and people components shape the outcome component and vice versa [39]. All three components can be observed from the organizational and individual perspective [39]. This distinction between task, social, cultural, and physical contexts of WE described by Lee, Stone [4] resonates in the organization and people component of the sociotechnical system. The outcome component added by Harrison, Henriksen [39] concerns quality of care and employee outcomes. The 36 elements we included cover all three components, including outcome-related elements such as 'job satisfaction' and 'patient-centered culture'.

Finally, when basic physical needs such as 'a good building', 'running water', and 'lighting' are available, more attention is given to the task, social, and cultural contexts of WE [40]. Our Delphi list contained only two physical context elements of WE, which were derived from our literature review and interviews with participants from Western Europe and North America.

When measuring an employees' experience of WE, it is important to determine in advance which elements are measured by preference and from which perspective. One needs to choose a psychometrically valid instrument that best fits these elements. We found nine WE measurement tools that include between 11 and 14 of the elements that we identified [15, 24–31]. However, measuring all elements with one instrument will probably result in a long, user-unfriendly questionnaire. It would be interesting to study how we can blend several instruments to cover all elements. However, not every measurement tool has proven to be

psychometrically valid and reliable [18]. It is wise to opt for a short, sound instrument that functions as a thermometer and identifies the WE areas that have issues. This could be followed by zooming in on the problem area with a specific in-depth instrument or qualitative method. Finally, achieving a positive WE is a responsibility shared by all members of a team, including management. Therefore, it is important to regularly discuss WE experiences with all team members to come to mutual understanding and create improvement initiatives.

## Limitations

Some limitations of this study warrant consideration. First, the response rate decreased between Delphi rounds 1 and 2 by 22%, despite two reminders emails sent during each round. Attrition of participants is a common phenomenon with the Delphi methodology [41] and influences the results. Nevertheless, the samples in all three rounds can be considered comparable. Second, this Delphi study included only experts from Western Europe and North America. This may have caused selection bias, since perceptions of WE are context-driven [16, 40]. Some caution with generalization is therefore recommended. Third, we chose a Delphi model with three pre-defined rounds. Although this is a known form of consensus establishment [20, 41], it might have led to the premature inclusion or exclusion of elements. It remains unclear what could have happened if a fourth round had been held. Fourth, the use of a 10-point rating scale may have led to some bias due to the risk of variation in interpretation by the participants. Nevertheless, we consider this 10-point rating scale as the best option for our context and research sample.

## Conclusions

This research has refined the broad description of WE–as the inner setting of the organization where staff interplay with the organization within which they work [4, 9]–by conducting a literature review, interviews with experts, and a Delphi study. We found 36 elements that were considered relevant for a positive WE and believe that a positive WE measurement tool should include these 36 elements. However, none of the current WE measurement tools include all 36 elements. It might be interesting to further develop or integrate existing WE measurement tools to measure all the elements.

A positive WE is important for providing optimal patient care and attracting and retaining healthcare professionals. Measuring the WE can help healthcare management to improve negative WEs. However, how or with which interventions this can be done is not clear yet. The results of this study enable decision-making for a measurement tool. However, it is important to consider different perspectives when measuring and improving the WE.

## Supporting information

**S1 Data. Overview elements from literature including complete reference list of the included papers.**
(XLSX)

**S2 Data. Overview elements emerged from interviews.**
(XLSX)

## Acknowledgments

The authors thank all participants of the interviews and Delphi rounds for their willingness to contribute and share their views.

## Author Contributions

**Conceptualization:** Susanne M. Maassen, Catharina van Oostveen, Hester Vermeulen, Anne Marie Weggelaar.

**Formal analysis:** Susanne M. Maassen, Catharina van Oostveen.

**Funding acquisition:** Susanne M. Maassen, Catharina van Oostveen, Hester Vermeulen, Anne Marie Weggelaar.

**Investigation:** Susanne M. Maassen.

**Methodology:** Susanne M. Maassen, Catharina van Oostveen, Hester Vermeulen, Anne Marie Weggelaar.

**Supervision:** Catharina van Oostveen, Hester Vermeulen, Anne Marie Weggelaar.

**Writing – original draft:** Susanne M. Maassen, Anne Marie Weggelaar.

**Writing – review & editing:** Catharina van Oostveen, Anne Marie Weggelaar.

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
