## [Decision Letter · Decision Letter 0]

22 Dec 2020

PONE-D-20-29509

Defining a positive work environment for hospital healthcare professionals: a Delphi study

PLOS ONE

Dear Dr. Maassen,

Thank you for submitting your manuscript to PLOS ONE. After careful consideration, we feel that it has merit but does not fully meet PLOS ONE’s publication criteria as it currently stands. Therefore, we invite you to submit a revised version of the manuscript that addresses the points raised during the review process.

Based on the recommendation of the reviewers, this manuscript is in a good shape to be accepted for publication pending some quick reviews for language, style and most importantly further elaboration on the methods especially the approach to reaching consensus using Delphi.

We look forward to receiving your revised manuscript.

Kind regards,

Mohamad Alameddine, MPH, Ph.D.

Academic Editor

PLOS ONE

Journal Requirements:

Additional Editor Comments (if provided):

Based on the recommendation of the reviewers, this manuscript is in a good shape to be accepted for publication pending some quick reviews for language, journal style and most importantly further elaboration on the methods especially the approach to reaching consensus using Delphi.

Reviewers' comments:

Reviewer's Responses to Questions

**Comments to the Author**

1. Is the manuscript technically sound, and do the data support the conclusions?

Reviewer #1: Yes

Reviewer #2: Yes

2. Has the statistical analysis been performed appropriately and rigorously? 

Reviewer #1: N/A

Reviewer #2: N/A

3. Have the authors made all data underlying the findings in their manuscript fully available?

Reviewer #1: Yes

Reviewer #2: Yes

4. Is the manuscript presented in an intelligible fashion and written in standard English?

Reviewer #1: Yes

Reviewer #2: Yes

5. Review Comments to the Author

Reviewer #1: I enjoyed reviewing this paper. I agree with the 3-round (versus only 2) for the Delphi study.

The authors did a good job with the literature review, covering several databases and while not specifically a systematic review, did include additional references/concepts identified in the sample articles. I am curious about the positive WE dependent variable (for hospitals), as related to timeframes (years, decades, etc) - which may be different among countries, policies, and hospital regulations/etc. It is my opinion that the validity of the study was further increased by interviewing a variety of hospital industry professionals at different leadership/management/position levels.

Most impressive is Table 4, as related/explained with the interview results at the end of the paper. Circling back to the task, social, cultural, and physical attributes (line 270) was also impressive.

Limitations are valid points, yet not of concern to me in the end, beyond the locale/sampling bias mentioned. The paper does a good job reviewing hospital WE at an overall level and has spurred ideas for future, more specific research topics/questions.

Reviewer #2: This article builds on previous a systematic review. The authors concluded the need for a standardized tool that could have been constructed based on their literature review

Needs some editing to ensure coherence and perhaps more elaboration on the Methods especially the approach to reaching consensus using Delphi.

6. PLOS authors have the option to publish the peer review history of their article (what does this mean?). If published, this will include your full peer review and any attached files.

Reviewer #1: No

Reviewer #2: No

---

## [Author Response · Author response to Decision Letter 0]

2 Feb 2021

Dear Professor Alameddine,

We are pleased to learn that PLOS ONE is interested in publishing our paper, “Defining a positive work environment for hospital healthcare professionals: a Delphi study”. We wish to thank you and both reviewers for your positive feedback and constructive suggestions to improve our manuscript. We have now revised our manuscript as suggested by the reviewers. Below, we provide a point-by-point reaction to the comments of each reviewer.

Feedback academic editor

Based on the recommendation of the reviewers, this manuscript is in a good shape to be accepted for publication pending some quick reviews for language, style and most importantly further elaboration on the methods especially the approach to reaching consensus using Delphi.

 We thank the academic editor for their positive reaction. A native-speaking English language editor has made stylistic changes to improve the clarity and readability of the text. However, this led to changes throughout the whole document and is visible in the track change version of the manuscript. In addition, we have adapted the paper so it conforms to the style requirements of PLOS ONE. 

 We acknowledge the editor and reviewer’s point concerning the approach to consensus in our Delphi study. We have responded to this issue in detail in our response to reviewer 2. 

Feedback reviewer 1

I enjoyed reviewing this paper. 

 We thank reviewer 1 for the positive reaction and compliments. 

I agree with the 3-round (versus only 2) for the Delphi study.

We thank reviewer 1 for this feedback and support for our choice of a three round Delphi design.

The authors did a good job with the literature review, covering several databases and while not specifically a systematic review, did include additional references/concepts identified in the sample articles. I am curious about the positive WE dependent variable (for hospitals), as related to timeframes (years, decades, etc) - which may be different among countries, policies, and hospital regulations/etc. 

This literature review was published separately [1]. In this review, we screened 37 papers on development and psychometric validation of work environment measurement tools. We also elaborated on the differences in time frames and countries. We specifically looked for instruments that measure hospital employees’ experiences of WE in the broadest sense because employees nowadays work together in interdisciplinary teams. However, in the literature and in practice, we noticed that certain aspects of WE require more attention from nurses than doctors or vice versa. In the Delphi paper, we refer to this publication in the Methods section, but not in the Results because only the results of the present study should be described in this section. To clarify this point, we have modified line 175 (in the previous version line 182) and have added the reference. 

It is my opinion that the validity of the study was further increased by interviewing a variety of hospital industry professionals at different leadership/management/position levels. 

Most impressive is Table 4, as related/explained with the interview results at the end of the paper. Circling back to the task, social, cultural, and physical attributes (line 270) was also impressive.

Limitations are valid points, yet not of concern to me in the end, beyond the locale/sampling bias mentioned. The paper does a good job reviewing hospital WE at an overall level and has spurred ideas for future, more specific research topics/questions.

We thank the reviewer for these compliments and positive feedback. We acknowledge that there is local/sampling bias. Unfortunately, hospital facilities are not the same all over the world, just as employee facilities. This will most likely lead to differences in the perception of the work environment. Because of these differences, we chose to delineate a specific selection. This made it possible to set up a practice-orientated study with input from hospital employees. The results are recognizable and applicable in practice. However, our results may not be generalizable to hospitals all over the world.

Feedback reviewer 2 

This article builds on previous a systematic review. The authors concluded the need for a standardized tool that could have been constructed based on their literature review

Needs some editing to ensure coherence and perhaps more elaboration on the Methods especially the approach to reaching consensus using Delphi.

 We thank reviewer 2 for the positive reaction and constructive feedback. We have addressed the coherence issues in cooperation with our native-speaking English language editor. Stylistic changes have been made throughout to improve the clarity and readability of the text.

In this Delphi study, we defined consensus as the percentage agreement with the statement ‘element belongs to a positive WE’. Delphi participants were asked to rate, on a 10-point scale (where 1 = ‘not at all’ and 10 = ‘totally’), to which extent an element belonged to the concept of positive WE. We chose a 10-point rating scale because this is a well-known rating scale for most Delphi participants and is commonly used in Delphi studies. Lange et al (2020) described a large variety in rating scales applied in Delphi studies [2]. There is no consensus on which type of rating scale is preferable. Lange et al. (2020) stated that the research question and context of each study determines which rating scale is suitable [2].

In our population, scores between 8 and 10 were considered ‘good’, scores between 5 and 7 ‘adequate’, and scores of 4 or lower ‘bad’. We counted the frequency of each rating on the 10-point scale, then applied two thresholds. For rounds 1 and 2, we set the threshold for inclusion at 80%, indicating that >80% of participants rated the element eight or higher. For round 3, we set the threshold at 70%, indicating that >70% of respondents rated the element as 8 or higher. 

We acknowledge that a 10-point rating scale may have led to some bias due to variation in interpretation by the participants. Nevertheless, for our context and research sample, we consider this the best option. We have added this point to the limitations section in line 305-308: ‘fourth, the use of a 10-point rating scale may have led to some bias due to the risk of variation in interpretation by the participants. Nevertheless, we consider this 10-point rating scale as the best option for our context and research sample.’

The consensus method is described in the ‘data collection’ and ‘data analysis and consensus’ sections. We have now referred the reader to the ‘data analysis and consensus’ section in line 154-156: ‘in the second round, all elements with consensus in the first round following the forward set threshold (see section ‘Data analysis & consensus’ for consensus method and thresholds).’ 

We also defended our choice of a 10-point rating scale in lines 151–153: ‘the 10-point rating scale is a commonly known rating scale for the Delphi participants and widely used in Delphi studies. A score of eight, nine or ten was considered as agreement.’ 

We hope that, with these revisions, you will now find our manuscript suitable for publication in PLOS ONE.

Sincerely, on behalf of all co-authors,

---

## [Editor Report · Decision Letter 1]

9 Feb 2021

Defining a positive work environment for hospital healthcare professionals: a Delphi study

PONE-D-20-29509R1

Dear Dr. Maassen,

We’re pleased to inform you that your manuscript has been judged scientifically suitable for publication and will be formally accepted for publication once it meets all outstanding technical requirements.

Kind regards,

Mohamad Alameddine, MPH, Ph.D.

Academic Editor

PLOS ONE

Additional Editor Comments (optional):

We thank the authors for addressing the comments made by the editor and reviewers. We have no additional comments.
---

## [Editor Report · Acceptance letter]

15 Feb 2021

PONE-D-20-29509R1 

Defining a positive work environment for hospital healthcare professionals: a Delphi study 

Dear Dr. Maassen:

I'm pleased to inform you that your manuscript has been deemed suitable for publication in PLOS ONE. Congratulations! Your manuscript is now with our production department. 

Kind regards, 

on behalf of

Dr. Mohamad Alameddine 

Academic Editor

PLOS ONE